# Roles of Noncoding RNAs in Regulation of Mitochondrial Electron Transport Chain and Oxidative Phosphorylation

**DOI:** 10.3390/ijms24119414

**Published:** 2023-05-28

**Authors:** Ami Kobayashi, Toshihiko Takeiwa, Kazuhiro Ikeda, Satoshi Inoue

**Affiliations:** 1Department of Neurology, Brigham and Women’s Hospital, Harvard Medical School, 60 Fenwood Rd., Boston, MA 02115, USA; akobayashi1@bwh.harvard.edu; 2Department of Systems Aging Science and Medicine, Tokyo Metropolitan Institute for Geriatrics and Gerontology, Itabashi-ku, Tokyo 173-0015, Japan; ttakeiwa@tmig.or.jp; 3Division of Systems Medicine & Gene Therapy, Saitama Medical University, Hidaka 350-1241, Japan; ikeda@saitama-med.ac.jp

**Keywords:** mitochondria, electron transport chain complex, electron transport chain supercomplex, oxidative phosphorylation (OXPHOS), noncoding RNAs

## Abstract

The mitochondrial electron transport chain (ETC) plays an essential role in energy production by inducing oxidative phosphorylation (OXPHOS) to drive numerous biochemical processes in eukaryotic cells. Disorders of ETC and OXPHOS systems are associated with mitochondria- and metabolism-related diseases, including cancers; thus, a comprehensive understanding of the regulatory mechanisms of ETC and OXPHOS systems is required. Recent studies have indicated that noncoding RNAs (ncRNAs) play key roles in mitochondrial functions; in particular, some ncRNAs have been shown to modulate ETC and OXPHOS systems. In this review, we introduce the emerging roles of ncRNAs, including microRNAs (miRNAs), transfer-RNA-derived fragments (tRFs), long ncRNAs (lncRNAs), and circular RNAs (circRNAs), in the mitochondrial ETC and OXPHOS regulation.

## 1. Introduction

Mitochondria play a crucial role in a diverse range of biological processes in eukaryotic cells. One of the important functions of mitochondria is to generate energy in the cell via electron transport and oxidative phosphorylation (OXPHOS). The mitochondrial electron transport chain (ETC) consists of a series of electron carriers embedded in the inner mitochondrial membrane. Electrons from NADH and FADH_2_, which are produced in the metabolic pathways of glucose and other nutrients, are transferred along the ETC, leading to the transport of protons across the membrane from the mitochondrial matrix to the intermembrane space. This establishes a proton gradient across the inner mitochondrial membrane, which is then used by the ATP synthase enzyme to produce ATP via OXPHOS. OXPHOS is the process of phosphorylating ADP to form ATP using energy from the proton gradient established by the ETC. The ETC comprises a series of four enzymatic complexes that contain complex I (NADH ubiquinone oxidoreductase/NADH dehydrogenase), complex II (succinate ubiquinone oxidoreductase/succinate dehydrogenase), complex III (ubiquinol cytochrome *c* oxidoreductase/cytochrome *bc*1 complex), and complex IV (cytochrome *c* oxidase). OXPHOS is accomplished by complex V (ATP synthase) [1]. Some of these complexes generate higher-order assemblies called ‘supercomplexes’ [1,2] (Figure 1). In mammalian cells, a supercomplex called ‘respirasome’ is the major structure, which is composed of one complex I, two complex IIIs, and one complex IV (I + III_2_ + IV). Although the biological function and regulation of the mitochondrial supercomplex assembly is highly debated, it can contribute to efficient energy production [3,4,5,6], reducing reactive oxygen species (ROS) [7,8,9], and the stability of complex I [10]. The ETC complexes and complex V are composed of specific protein subunits that are encoded in nuclear DNA (nDNA) or mitochondrial DNA (mtDNA) [11]. Some mutations of these subunits as well as assembly factors of ETC complexes and complex V are known to cause the dysregulation of mitochondrial functions associated with mitochondrial diseases such as mitochondrial encephalopathy, resulting in lactic acidosis and stroke-like episodes (MELAS), and Leigh syndrome. Furthermore, ETC/OXPHOS disorders are assumed to result in metabolism-related diseases and cancers [12,13,14,15,16]. Thus, a comprehensive understanding of ETC/OXPHOS regulation is crucial for precise diagnosis of and therapeutic intervention in mitochondria- and metabolism-related diseases and cancers.

Recent studies, including next-generation sequencing strategies, have reported that noncoding RNAs (ncRNAs) play key roles in mitochondrial functions, such as the ETC and OXPHOS, through various biological processes, including epigenetic and post-transcriptional gene regulation, and are involved in mitochondria- and metabolism-related diseases. For example, among microRNAs (miRNAs), a type of noncoding RNA, *miR-181c*, has been reported to be associated with reduced exercise capacity and cardiac dysfunction via inhibiting the translation of *cytochrome c oxidase subunit 1* (*COX1*) and reducing complex IV’s function [17]. In another type of ncRNA, namely, long ncRNAs (lncRNAs), *metastasis-associated lung adenocarcinoma transcript 1* (*MALAT1*) has been suggested to promote hepatocellular carcinoma (HCC) cell proliferation, migration, and invasion by epigenetically regulating mtDNA-encoded ETC-related genes [18]. These ncRNAs may be promising targets in the diagnosis and treatment of related diseases [19,20].

In this review, we summarize the functional roles of ncRNAs, including miRNAs, transfer RNA-derived fragments (tRFs), lncRNAs, and circular RNAs (circRNAs), in the cross-talk signaling between mitochondria and the nucleus targeting regulated ETC complexes in mammalian cells.

## 2. Species and Classification of ncRNAs Affecting Mitochondria–Nucleus Crosstalk

In mammalian cells, numerous ncRNAs are synthesized to exhibit their various functions, including RNA splicing, RNA modification, and mRNA translation. ncRNAs are classified according to their length and structures. Small ncRNAs (sncRNAs) are classified as ncRNAs that are shorter than 200 nucleotides. miRNAs are a class of sncRNAs that are approximately 22 nucleotides long. miRNAs are synthesized as primary transcripts in the nucleus and are subsequently cleaved by the DROSHA/DiGeorge syndrome critical region 8 (DGCR8) complex to form single stem-loop precursor molecules, which are known as pre-miRNAs. Pre-miRNAs are generally transported to the cytoplasm by a nuclear transport receptor, Exportin 5, along with a cofactor RanGTP, and are then further cleaved by DICER1 to generate mature double-stranded miRNA duplexes [21]. In most cases, the miRNA duplex is incorporated into the RNA-binding protein argonaute 2 (AGO2) in mammalian cells; then, only one RNA strand (the guide strand) forms a complex with AGO2, leading to the formation of the miRNA-induced silencing complex (miRISC). The miRISC typically recognizes the 3′ untranslated region (3′ UTR) of target mRNAs using the guide strand and downregulates the translation and stability of these mRNAs. [22,23,24,25]. Besides their role in negative post-transcriptional regulation, miRNAs have been proposed to possess non-canonical functions outside the cytoplasm. For example, some miRNAs may act as chromatin and transcriptional regulators in the nucleus and as translational activators in mitochondria [26,27]. Mitochondrially located miRNAs are commonly termed “mitomiRs”, which are derived from either transcripts of mitochondria or the nuclear genome [28,29]. Nuclear genome-encoded mitomiRs are more abundant in mitochondria compared to mitochondrial genome-encoded genes. The nuclear genome-encoded mitomiRs are often located in loci within or close to mitochondrial genes so that their transcriptions can be coregulated.

tRNA derived fragments (tRFs) are another class of sncRNAs, which are approximately 16–35 nt in length and originate from the cleavage of precursor or mature tRNAs by specific nucleases under certain conditions including cellular stress [30,31,32,33,34,35,36,37]. tRFs can be derived from tRNAs found in both the nucleus and mitochondria [38,39,40,41,42,43]. In the human genome, more than 400 nuclear genes encode for approximately 280 different tRNA isotypes, and mitochondrial DNA encodes 22 tRNAs [44,45,46]. The diversity of tRNA molecules can be attributed to the presence of isoacceptors, which are tRNAs that carry the same amino acid but have different nucleotide sequences in both the body and the anticodon region, as well as isodecoders, which are tRNAs that possess the same anticodon and carry the same amino acid but differ in their nucleotide sequences in the body. Isoacceptors and isodecoders exist in several copies in the genome and their expressions are tissue- and cell-type-specific, which can be modulated by stress conditions. Post-transcriptional tRNA modifications can take place in the nucleus, cytosol, and mitochondria and are induced by various tRNA-modifying enzymes through modulating base pairing, tRNA folding, and stability [47]. On average, cytoplasmic and mitochondrial tRNAs in mammals contain 13 and 5 modified bases, respectively. Hypomodified tRNAs are targeted for degradation and undergo endonuclease-mediated fragmentation. tRFs resulting from the fragmentation of mature tRNAs are classified in six major structural categories based on their positions in the parental tRNA sequence, as follows: 5′-tRFs and 3′-tRFs generated from either distal end of mature tRNAs, internal tRFs (i-tRFs) from internal sequences of mature tRNAs, 5′-tRNA halves and 3′-tRNA halves from the cleavage of anticodon sites of mature tRNAs (tRNA-derived stress-induced RNA (tiRNA)-5s and -3s, respectively), and tRF-1s generated from precursor tRNAs. While angiogenin is the enzyme primarily responsible for cleaving the anticodon site, different endonucleases such as DICER, SLNF13, RNase T2, and RNase Z have been reported to generate other classes of tRFs [48,49,50]. tRF is often stimulated under stress conditions, in which post-transcriptional modifications of tRNA are impaired and function to repress gene expression in a manner similar to that of miRNAs [51,52]. Moreover, tRFs function as important modulators of many aspects of biological processes, including translation, apoptosis, immunity, nutritional uptake, and oxidative stress. Notably, with the innovative development of high-throughput sequencing techniques and bioinformatic analysis programs, it has become evident that nuclear genome-encoded tRNAs and mitochondrial tRNAs (mt-tRNAs) can produce tRFs [43,53,54,55]. mt-tRNA-derived tRFs (mt-tRFs) actively participate in intracellular communication and mitochondrial pathophysiology [55,56,57].

Noncoding RNAs longer than 200 nucleotides are referred to as lncRNAs. LncRNAs are transcribed from both nuclear and mitochondrial genomes, and there are approximately 20,000 lncRNA genes in humans, showing a diversity comparable to that of protein-coding genes. Although most lncRNAs remain functionally unannotated, some lncRNAs are thought to play essential roles in cell physiology and numerous biological processes and diseases, including various types of cancers [20,58,59,60,61]. Some nuclear genome-encoded lncRNAs have been found in mitochondria. For example, one nuclear genome-encoded lncRNA, namely, *RNA component of mitochondrial RNA processing endoribonuclease* (*RMRP*), is imported to the mitochondrial matrix [62]. This lncRNA was identified as a component of the mitochondrial RNA-processing endoribonuclease (MRP) and has been suggested to be involved in the generation of primers for mitochondrial DNA replication [63]. Meanwhile, the role of *RMRP* in mitochondria is debated, as recent studies have suggested that RNA primer generation is mediated by the premature arrest of RNA polymerase mitochondrial (POLRMT), and the 3′ half of this lncRNA (~130 nt) has been predominantly detected inside mitochondria [64,65].

CircRNAs are defined by their structure and not by their length. Such structures are single-stranded and covalently closed at the 5′ and 3′ ends [66,67]. The production of circRNA generally occurs through a distinct form of alternative splicing referred to as back-splicing. This process involves the 3′ end of an exon joining to the 5′ end of its own or a preceding exon through a 3′,5′-phosphodiester bond, resulting in a circular structure with a back-splicing junction site [68,69,70]. The unique structure of circRNAs provides them with a longer-half life and greater resistance to exonucleases than linear RNAs [68,69,71,72,73]. An increasing number of investigations uncovered diverse roles of circRNAs, such as acting as protein scaffolds or miRNA sponges, as well as having the ability to undergo translation into polypeptides [69,74,75]. Recently, mitochondria-located circRNAs have been discovered, and their functions have been investigated, which has expanded our understanding of circRNA derivation and mitochondrial transcriptome [76,77,78].

Regarding the targeting and import of nDNA-encoded ncRNAs into mammalian mitochondria, some proteins have been indicated as being involved in these processes. For example, polynucleotide phosphorylase (PNPASE) has been suggested to modulate the import of some ncRNAs, including *RNase P* RNA and *RMRP*. However, the mechanisms whereby nDNA-encoded ncRNAs target and are imported into mitochondria remain controversial, and thus new techniques are needed to precisely analyze nuclear-mitochondrial RNA dynamics (reviewed in [64]).

In this review, we define mitochondrial ncRNAs (mt-ncRNAs) as ncRNAs that are either transcribed from mtDNA regardless of subsequent mitochondrial localization or are mitochondria-located ncRNAs transcribed from nDNA. Another classification for ncRNAs that are neither located in mitochondria nor encoded by the mitochondrial genome but regulate mitochondrial functions is “mitochondria-associated ncRNAs”. Mitochondria-associated ncRNAs, a large portion of which are miRNAs, regulate mitochondrial functions by targeting nuclear genome-encoded mitochondrial mRNAs in the cytoplasm (Table 1, Figure 2). It has been suggested that mt-ncRNAs and mitochondria-associated noncoding RNAs regulate mitochondria–nucleus crosstalk through anterograde and retrograde signals [79,80].

## 3. Roles of microRNAs in the Regulation of Mitochondrial ETC Complexes and OXPHOS

Numerous mitomiRs have been shown to affect the levels of enzymes encoded by the mitochondrial genome in ETC complexes (Table 2 and Figure 3). For instance, the mitochondria-located miRNA *let-7a* in MCF-7 breast cancer cells decreases the activity of ETC complex I by inducing the destabilization of the mRNA encoding NADH ubiquinone oxidoreductase core subunit 4 (ND4), which is a subunit of complex I [81].

It has been suggested that *miR-762* translocates into mitochondria in murine cardiomyocytes upon anoxia/reoxygenation treatment and decreases complex I activity and intracellular ATP levels, increases ROS levels, and enhances apoptotic cell death through suppressing the expression of NADH ubiquinone oxidoreductase chain 2 (ND2), which is a subunit of complex I [82]. Intriguingly, the administration of an inhibitor (antagomir) of *miR-762* ameliorated myocardial ischemia/reperfusion injury in mice. Another example of the negative regulation of ETC complexes is provided by *miR-181c*, which binds to the 3′ UTR of *cytochrome c oxidase subunit 1* (*COX1*) mRNA, encoding a subunit of complex IV and suppressing *COX1* translation in rat cardiac myocytes. The overexpression of *miR-181c* dramatically decreased COX1 protein levels, leading to complex IV remodeling and altering mitochondrial function, including by precipitating augmented ROS production [17]. Notably, the systemic administration of *miR-181c* in rats diminished exercise capacity and caused heart failure [101]. For patients with heart failure with a preserved ejection fraction (HFpEF), exercise training is a treatment that commonly used to improve such patients’ aerobic capacity and quality of life. A recent study has shown that the circulating level of *miR-181c* is significantly higher in low responders to exercise training, suggesting that *miR-181c* may affect the exercise adaptation process in HFpEF patients partly through ETC regulation [102].

MitomiRs also negatively regulate ATP production by targeting complex V subunits. In a mouse model of type I diabetes mellitus, it has been shown that *miR-378a* redistributes to interfibrillar mitochondria (IFM) and downregulates the expression of *ATP synthase F0 subunit 6* (*ATP6*), which encodes a subunit of complex V. The administration of an inhibitor of *miR-378a* alleviated ATP6 expression in IFM and heart failure in this mouse model [83]. In addition, *miR-378a* levels increased in subsarcolemmal mitochondria (SSM) but not in IFM, and *miR-378a* was responsible for modulating ATP6 expression in a mouse model of type II diabetes mellitus (*db*/*db* mice) [84]. Intriguingly, the knockout (KO) of *miR-378a* rescued complex V activity and cardiac function in *db*/*db* mice, thus suggesting that *miR-378a* may be a potential therapeutic target for treating diabetes mellitus. Further study on the molecular mechanism by which *miR-378a* targets mitochondria will be important for its clinical application [97].

For the positive regulation of ETC complexes, miR-21, a mitomiR, interacts with the open reading frame of the mRNA of *cytochrome b* (*CYTB*), encoding a subunit of complex III, and promotes the translation of *CYTB* to reduce ROS production. Experiments using spontaneous hypertensive rats have suggested that this mitomiR reduces blood pressure and suppresses cardiac hypertrophy [85]. *MiR-1* is another example of a mitomiR that enhances the mitochondrial translation of ETC complexes. The muscle-specific *miR-1*, which is induced during myogenesis, directly and specifically recruits mRNAs encoding NADH ubiquinone oxidoreductase core subunit 1 (ND1), a subunit of complex I, and COX1 to mitochondrial ribosomes and promotes the translation of these mRNAs. This function of miR-1 is dependent on AGO2 but is independent of an AGO2 partner glycine-tryptophan protein of 182 kDa (GW182) [86]. Another mitomiR, *miR-5787*, promotes the translation of *cytochrome c oxidase subunit 3* (*COX3*) in tongue squamous cell carcinoma (TSCC) cells. In Cal27 and Scc25 TSCC cells, the decrease in *miR-5787* expression contributes to the shifting of glucose metabolism from OXPHOS to glycolysis and enhances cisplatin resistance [87].

It has also been demonstrated that several mitomiRs affect ETC complexes via regulating mitochondrial gene expression through a route that does not occur post-transcription. The mitomiR *miR-2392* has been reported to directly regulate mtDNA transcription. *miR-2392* recognizes target sequences in the H-strand and partially represses polycistronic mtDNA transcription in a cell-type-specific fashion. As a consequence, *miR-2392* represses the expression of ETC complex subunits, such as ND4, COX1, and cytochrome *c* oxidase subunit 2 (COX2), to enhance anaerobic respiration in TSCC cells [88].

Mitochondria-associated miRNAs that are located in the cytoplasm modulate mitochondrial functions by targeting nuclear genome-encoded mitochondrial transcripts in the cytoplasm (Table 2 and Figure 3). Some studies have reported that miRNAs affect ATP synthesis by altering the expression of the ETC complex and complex V subunits. For instance, one brain-specific miRNA, *miR-338*, lowers the expression of the nuclear genome-encoded cytochrome *c* oxidase subunit 4 (COX4), a subunit of complex IV, after its binding to the 3′-UTR of its mRNA [89]. When an *miR-338* inhibitor was transfected into axons of neural cells, metabolic oxygen consumption was increased by about 50% when compared with nontargeting cells. Other examples include *miR-210*, which modulates mitochondrial functions during hypoxia by directly reducing the expression levels of cytochrome *c* oxidase assembly factor heme A:farnesyltransferase COX10, which facilitates the assembly of complex IV, and *miR-101*, which inhibits the replication of herpes simplex virus-1 (HSV-1) by downregulating the expression of ATP synthase F1 subunit β (ATP5B), a subunit of complex V [90,91]. *miR-663* has been shown to increase the expression levels of ETC complex (I, II, III, and IV) subunits and assembly factors, thereby stabilizing the ETC supercomplexes. Authors have proposed that *miR-663* may exert tumor-suppressive effects in breast cancer through enhancing ETC and OXPHOS activity [92]. Although ubiquinol cytochrome *c* reductase complex assembly factor 2 (UQCC2) is a potential target of *miR-663*, the mechanism by which *miR-663* controls the expression of a wide range of ETC-related proteins remains unclear.

Interestingly, recent studies show the involvement of miRNAs in maladaptive retrograde signaling in response to OXPHOS dysfunction caused by pathogenic mtDNA mutations [103,104,105]. Dysfunction in OXPHOS caused by various pathological mutations modulates the levels of ROS-sensitive miRNAs. These miRNAs regulate the expression of mitochondrial tRNA-modifying enzymes [103,104]. By regulating the modification of mt-tRNAs, these miRNAs can potentially affect the speed and accuracy of mitochondrial translation, which can help mitigate the stoichiometric imbalance between mtDNA- and nuclear genome-encoded ETC complexes and complex V subunits originated through pathological mtDNA mutation. In addition, pathological mtDNA mutation can lead to the dysregulation of other miRNAs, which significantly affects the nuclear expression and potentially explains certain characteristics of mtDNA disease [106].

## 4. Roles of tRNA-Derived Small Fragments in the Regulation of Mitochondrial ETC Complexes

Since many mitochondrial diseases are caused by mutations in mt-tRNA genes, it follows that mitochondrial translation is involved in the pathophysiology of cells and tissues. Human mtDNA is a circular and double-stranded DNA consisting of 16,569 base pairs that encodes 37 genes. Thirteen of these genes are translated through specific decoding via mitochondrial ribosomes to the essential subunits of mitochondrial ETC complexes I, III, IV, and also complex V [107]. Mutations in mitochondrial DNA, particularly in mt-tRNA, are associated with syndromes such as myoclonic red fiber epilepsy (MERRF) and cardiomyopathy [108].

As tRNA-derived small fragments (tRFs) have recently been discovered to be regulators of mitochondrial biology, there are only a few reports on the roles of tRFs in relation to ETC complex regulation (Table 2 and Figure 3). As previously mentioned, mtDNA mutations causing OXPHOS dysfunction lead to changes in the levels of certain miRNAs controlling the modification of mt-tRNAs. One well-known example is the m.3243A>G mutation in the *mitochondrially encoded tRNA* leucine *1* (*UUA*/*G*) (*MT-TL1*) gene that encodes *mt-tRNA^Leu(UUR)^*. It is one of the most common mitochondrial pathogenic mutations, with a carrier frequency estimated in the range between 0.95 and 18.4 per 100,000 people in northern European populations [109,110,111]. This mutation is associated with several diseases such as MELAS, chronic progressive external ophthalmoplegia (CPEO), diabetes mellitus, and deafness. Although the manifestation of diseases varies significantly in different tissues based on the proportion of mutant mtDNA molecules present, patients with an m.3243A>G mutation in the *MT-TL1* gene often show a severe, pre-existing deficiency in ETC complexes I and IV [112,113]. Accumulated data suggest that a deficiency in the aminoacylation of mutant *mt-tRNA^Leu(UUR)^* and a lack of taurine-containing modifications of its anticodon wobble position affecting the recognition of UUG codons (U34) may constitute the origin of a mitochondrial translation defect [114,115,116,117,118]. This leads to a decrease in steady-state levels of ETC complexes and an affected mitochondrial respiration rate. In addition, ROS-sensitive miRNAs, termed miR-9/9*, are induced by ETC complex deficiency, which downregulates the expression of U34-modification enzymes and leads to the hypomodification of several mt-tRNA species, including *mt-tRNA^Glu(UUC)^*. This causes the accumulation of *mt-i-tRF^Glu(UUC)^*, which is produced by the action of the nuclease DICER. An increase in *mt-i-tRF^Glu(UUC)^* downregulates the expression of the nuclear gene mitochondrial pyruvate carrier 1 (*MPC1*), promoting the increase in extracellular lactate levels and leading to lactic acidosis, which is one of the well-known phenotypes of MELAS [57]. This report was the first to demonstrate the involvement of tRF in the phenotype of ETC complex deficiency by clarifying the mitochondria–nucleus crosstalk controlled by small noncoding RNAs, while the hypomodification at important sites in mature tRNAs is assumed to be the underlying mechanism of the clinical manifestation of this disease with ETC complex deficiencies [119,120,121,122]. Furthermore, a recent study reported that 5′-tRNA halves derived from *tRNA^His(GTG)^*, *tRNA^Glu(CTC)^*, and *tRNA^Glu(TTC)^* (tiRNA-5^His(GTG)^, *tiRNA-5^Glu(CTC)^*, and *tiRNA-5^Glu(TTC)^*, respectively) modulate MPC1 expression in neonatal rat pancreatic β cells and that an inhibitor cocktail of these tRNA halves suppresses the mitochondrial respiration, proliferation, and insulin secretion of these cells, suggesting their importance in postnatal β cell maturation. *tiRNA-5^His(GTG)^* and *tiRNA-5^Glu(CTC)^* are assumed to mediate these functions through mitochondria since they were found in a mitochondrial fraction [93].

## 5. Roles of Long Noncoding RNAs in the Regulation of Mitochondrial ETC Complexes

mt-lncRNAs, including both RNAs derived from mtDNA and nuclear genome-encoded lncRNAs transported into mitochondria, collaborate with transcription factors and other epigenetic regulators to regulate mitochondrial gene expression and function. These mt-lncRNAs play a crucial role in multiple gene-regulatory networks, serving as possible epigenetic mediators that coordinate nuclear and mitochondrial functions [62,123] (Table 2 and Figure 3). One mt-lncRNA that is reported to regulate the expression of ETC complexes is *metastasis-associated lung adenocarcinoma transcript 1* (*MALAT1*). *MALAT1* is a nuclear genome-encoded lncRNA. A recent study indicated that *MALAT1* is located in mitochondria in HepG2 hepatocellular carcinoma (HCC) cells. RNA reverse transcription-associated trap sequencing (RAT-seq) and chromatin isolation via an RNA purification (ChIRP) assay showed that *MALAT1* interacts with multiple sites on mtDNA, including the D-loop region and the *COX2* and *NADH ubiquinone oxidoreductase core subunit 3* (ND3) gene loci. *MALAT1* modulates the CpG methylation of mtDNA and the expression of mtDNA-encoded ETC-related genes, such as *COX1*, *COX2*, *ND1*, and *ND3* genes. *MALAT1* depletion impairs mitochondrial biogenesis and intracellular ATP synthesis; suppresses HCC cell proliferation, migration, and invasion; and induces mitochondria-associated apoptosis. Thus, *MALAT1* may be an important therapeutic target for HCC, and the significance of *MALAT1*’s function in mitochondria requires further study [18].

Another nuclear-genome-encoded lncRNA, *survival-associated mitochondrial melanoma-specific oncogenic non-coding RNA* (*SAMMSON*), has been shown to localize in mitochondria in melanoma cells. In these cells, *SAMMSON* interacts with p32, a master regulator of mitochondrial homeostasis and metabolism, and facilitates the mitochondrial localization of p32. The silencing of *SAMMSON* decreases the expression levels of mtDNA-encoded ETC complex subunits and the activities of ETC complexes I and IV and suppresses tumorigenesis via enhancing apoptosis [94,95]. Meanwhile, a recent study showed that *SAMMSON* silencing increases the expression levels of ETC complexes, cellular respiration, and mitochondrial replication in doxorubicin-resistant MCF-7 cells [96]. This finding suggests context-dependent functions for SAMMSON, although further analysis is required to clarify the binding partners and molecular mechanism of this lncRNA.

In addition, it has been indicated that some mitochondria-associated lncRNAs function as miRNA sponges in order to modulate the expression of ETC complexes. For example, it has been suggested that the lncRNA *potassium voltage-gated channel subfamily Q member 1 overlapping transcript 1* (*Kcnq1ot1*) acts as a sponge for *miR-378a* and upregulates ATP6 expression and complex V activity in cardiomyocytes [97]. Another lncRNA, *cytoplasmic endogenous regulator of oxidative phosphorylation 1* (*Cerox1*), binds to and inhibits *miR-488-3p* to enhance the expression of multiple ETC complex I subunits and mitochondrial respiration in N2A neuroblastoma cells [98].

## 6. Roles of Circular RNAs in the Regulation of Mitochondrial ETC Complexes

Recent studies have indicated that circRNAs can act as scaffolds in protein complex assembly, and some circRNAs regulate mitochondrial functions through direct binding to subunits of ETC complexes and complex V (Table 2 and Figure 3). For instance, the mitochondria-located *circRNA circular RNA pumilio-1* (*circPUM1*) has been reported to act as a scaffold in order to regulate the spatial conformation of ETC complex III in KYSE30 and KYSE410 esophageal squamous cell carcinoma (ESCC) cells [99]. *circPUM1* binds to ubiquinol cytochrome *c* reductase core protein 2 (UQCRC2) and modulates the formation of the UQCRC1-UQCRC2 dimer in mitochondrial complex III, which helps pump hydrogen from the mitochondrial matrix to the mitochondrial intermembrane space [124]. The silencing of *circPUM1* leads to an intense interaction between UQCRC1 and UQCRC2, whereas *circPUM1* overexpression disrupts it. Furthermore, *circPUM1* silencing decreases ATP synthesis, whereas *circPUM1* overexpression promotes it. Conversely, ROS levels are increased through *circPUM1* silencing and decreased via *circPUM1* overexpression. These findings suggest that *circPUM1* plays a critical role in the proper assembly of the UQCRC1-UQCRC2 dimer and, therefore, OXPHOS. Intriguingly, *circPUM1* silencing induces pyroptosis, a kind of programmed cell death, suggesting that this circRNA is a potential target of ESCC treatment. Furthermore, it has been reported that a circRNA regulates mitochondrial ROS output through interacting with a complex V subunit, namely, ATP5B [76]. *Steatohepatitis-associated circRNA ATP5B regulator* (*SCAR*) is a mitochondrial genome-encoded circRNA that is downregulated in fibroblasts of liver tissues from patients with non-alcoholic steatohepatitis (NASH) cirrhosis. *SCAR* binds to ATP5B, which acts as a regulator of the mitochondrial permeability transition pore (mPTP) complex. The *SCAR*–ATP5B complex blocks mPTP from cyclophilin D, a facilitator of mPTP opening, and shuts down mPTP, thereby inhibiting mitochondrial ROS output. Interestingly, circRNA *SCAR* alleviates high fat diet-induced cirrhosis in mice, suggesting that the manipulation of circRNA *SCAR* expression may be applicable to therapeutic strategies of NASH.

Moreover, *mitochondrial circRNA for translocating phosphoglycerate kinase 1* (*mcPGK1*) has been indicated to be involved in metabolism reprogramming from OXPHOS to glycolysis. *mcPGK1* promotes the mitochondrial import of a key glycolytic enzyme, PGK1, by facilitating the interaction between PGK1 and the translocase of outer mitochondrial membrane 40 (TOM40), which forms the central channel of the translocase of the mitochondrial outer membrane (TOM complex). *mcPGK1* is highly expressed in primary liver tumor-initiating cells (TICs) and is involved in TIC self-renewal, indicating the significance of *mcPGK1* in clinical management of liver cancer [100].

## 7. Conclusions

In the present review, we have described multiple classes of noncoding RNAs and their regulatory pathways involved in the mitochondrial electron transport chain and OXHPOS (Table 2 and Figure 3). Clarifying the mechanisms of the noncoding RNAs regulating the mitochondrial electron transport chain and OXHPOS will help identify therapeutic targets for various diseases such as MERRF, MELAS, and cardiomyopathy in which OXPHOS plays a crucial role. The identification of new noncoding RNAs affecting the mitochondrial electron transport chain, OXPHOS, and their related molecules will allow us to further understand pathophysiology of the diseases. Although some noncoding RNAs mentioned in this review can be targets for innovative treatment or prophylactic agents for OXPHOS-related conditions, they are still far from clinical application. Both basic and clinical research on noncoding RNAs and the pathways by which they regulate the mitochondrial electron transport chain and OXPHOS would be required for the discovery of novel therapeutic approaches for OXPHOS-related diseases.

## Figures and Tables

**Figure 1 ijms-24-09414-f001:**
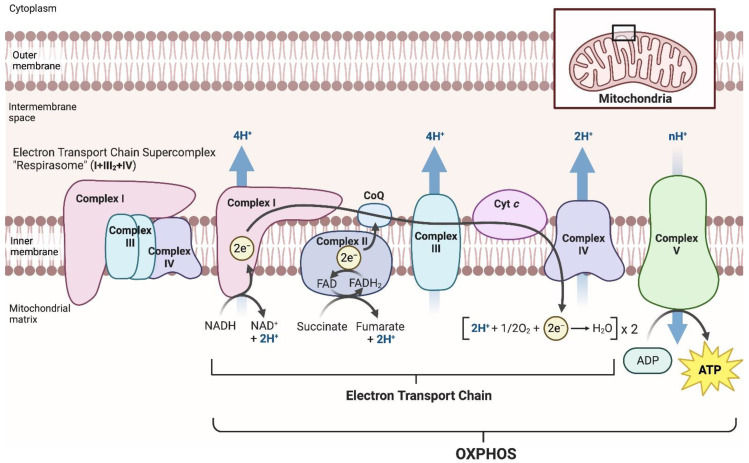
Mitochondrial electron transport chain complexes, supercomplex, and OXPHOS. The mitochondrial electron transport chain (ETC) is operated by four enzymatic protein complexes in the inner mitochondrial membrane, i.e., complex I (NADH ubiquinone oxidoreductase/NADH dehydrogenase; CI), complex II (succinate ubiquinone oxidoreductase/succinate dehydrogenase; CII), complex III (ubiquinol cytochrome *c* oxidoreductase/cytochrome *bc*_1_ complex; CIII), and complex IV (cytochrome *c* oxidase; CIV). Complexes I–IV mediate electron transport in the sequence from NADH or FADH_2_ to molecular oxygen. Through the ETC, protons (H^+^) are transported across the inner mitochondrial membrane to the intermembrane space. The established proton gradient is used to generate ATP from ADP phosphorylation by complex V (ATP synthase; CV), which is known as electron transport-linked phosphorylation or oxidative phosphorylation (OXPHOS). A certain portion of the mitochondrial electron transport complexes form higher-order structures called ‘supercomplexes’. The supercomplex generated by one CI, two CIIIs, and one CIV has been termed the ‘respirasome’.

**Figure 2 ijms-24-09414-f002:**
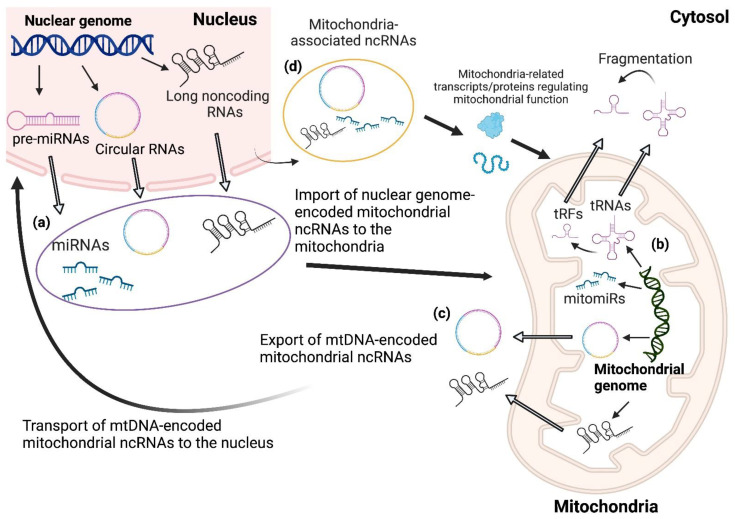
Noncoding RNAs regulating mitochondria–nucleus crosstalk. According to their localization and genome origin, mitochondrial noncoding RNAs (mt-ncRNAs) can be classified into four types: (**a**) nuclear-genome-encoded mitochondria-located mt-ncRNAs, (**b**) mtDNA-encoded mitochondria-located mt-ncRNAs, (**c**) mtDNA-encoded nucleus/cytosol-located mt-ncRNAs, and (**d**) nuclear-genome-encoded nucleus/cytosol-located mt-ncRNAs (termed “mitochondria-associated noncoding RNAs”). mt-ncRNAs regulate the mitochondrial electron transport chain and OXPHOS, which coordinate the crosstalk between nucleus and mitochondria as well as mitochondria-associated noncoding RNAs.

**Figure 3 ijms-24-09414-f003:**
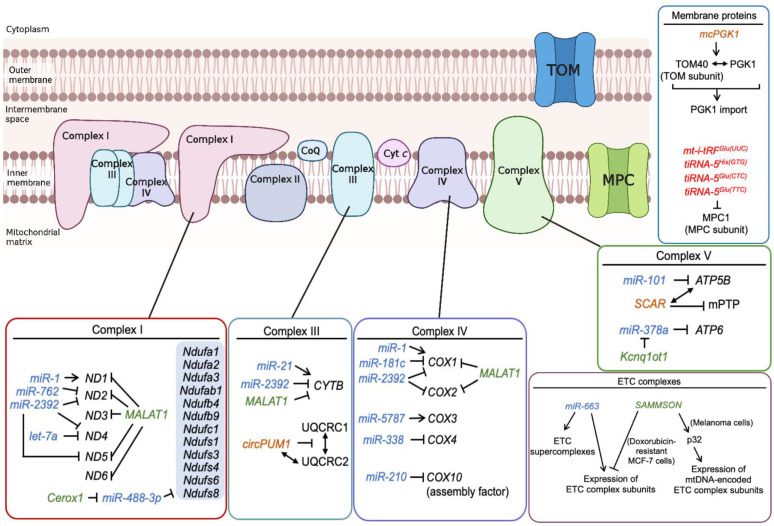
Regulation of mitochondrial ETC and OXPHOS by ncRNAs. The roles of the ncRNAs listed in Table 2 in mitochondrial ETC and OXPHOS are illustrated herein. miRNAs, tRFs, lncRNAs, and circRNAs are represented in blue, red, green, and orange, respectively. *Ndufa1*, *NADH ubiquinone oxidoreductase subunit A1*; *Ndufa2*, *NADH ubiquinone oxidoreductase subunit A2*; *Ndufa3*, *NADH ubiquinone oxidoreductase subunit A3*; *Ndufab1*, *NADH ubiquinone oxidoreductase subunit AB1*; *Ndufb4*, *NADH ubiquinone oxidoreductase subunit B4*; *Ndufb9*, *NADH ubiquinone oxidoreductase subunit B9*; *Ndufc1*, *NADH ubiquinone oxidoreductase subunit C1*; *Ndufs1*, *NADH ubiquinone oxidoreductase core subunit S1*; *Ndufs3*, *NADH ubiquinone oxidoreductase core subunit S3*; *Ndufs4*, *NADH ubiquinone oxidoreductase subunit S4*; *Ndufs6*, *NADH ubiquinone oxidoreductase subunit S6*; *Ndufs8*, *NADH ubiquinone oxidoreductase core subunit S8*; TOM, translocase of the mitochondrial outer membrane; TOM40, translocase of outer mitochondrial membrane 40.

**Table 1 ijms-24-09414-t001:** Classification of noncoding RNAs (ncRNAs) related to mitochondria.

Noncoding RNAs (ncRNAs) Related to Mitochondria	Definition	Genome	Location
**miRNA (~22 nt)**			
mt-miRNAs	mtDNA-encoded miRNAs or miRNAs located in mitochondria	Nuclear	Mitochondria
		Mitochondrial	Mitochondria or other compartments
MitomiRs	Mitochondria-located miRNAs	Nuclear or Mitochondrial	Mitochondria
Mitochondria-associated miRNAs	miRNAs that are neither located in mitochondria nor encoded by mtDNA but regulate mitochondrial functions	Nuclear	Not located in mitochondria
**tRNA-derived fragments (tRFs) (16–35 nt)**			
mt-tRFs	mtDNA-encoded tRFs or tRFs located in mitochondria	Nuclear	Mitochondria
		Mitochondrial	Mitochondria or other compartments
Mitochondria-associated tRFs	tRFs that are neither located in mitochondria nor encoded by mtDNA but regulate mitochondrial functions	Nuclear	Not located in mitochondria
**Long noncoding RNAs (lncRNAs) (>200 nt)**			
mt-lncRNAs	mtDNA-encoded lncRNAs or lncRNAs located in mitochondria	Nuclear	Mitochondria
		Mitochondrial	Mitochondria or other compartments
Mitochondria-associated lncRNAs	lncRNAs that are neither located in mitochondria nor encoded by mtDNA but regulate mitochondrial functions	Nuclear	Not located in mitochondria
**Circular RNAs (circRNAs)**			
mt-circRNAs	mtDNA-encoded cicRNAs or circRNAs located in mitochondria	Nuclear	Mitochondria
		Mitochondrial	Mitochondria or other compartments
Mitochondria-associated circRNAs	circRNAs that are neither located in mitochondria nor encoded by mtDNA but regulate mitochondrial functions	Nuclear	Not located in mitochondria

**Table 2 ijms-24-09414-t002:** Roles of ncRNAs in regulation of mitochondrial ETC and OXPHOS.

ncRNA	Species	Genome	Roles in Mitochondrial ETC and OXPHOS
**miRNAs**			
**mitomiRs**			
*let-7a*	Human	Nuclear	Decreases complex I activity by destabilizing *ND4* mRNA in MCF-7 breast cancer cells [81]
*miR-762*	Mouse	Nuclear	Decreases complex I activity through decreasing ND2 in murine cardiomyocytes upon anoxia/reoxygenation treatment [82]
*miR-181c*	Rat	Nuclear	Induces complex IV remodeling and increases ROS levels by suppressing *COX1* translation in rat cardiomyocytes [17]
*miR-378a*	Mouse	Nuclear	Decreases a complex V subunit, ATP6, in IFM of a mouse model of type I diabetes mellitus [83] Downregulates ATP6 expression in SSM of a mouse model of type II diabetes mellitus (*db*/*db* mice) [84]
*miR-21*	Rat	Nuclear	Promotes *CYTB* translation in H9c2 rat cardiomyocytes and in hearts of spontaneous hypertensive rats [85]
*miR-1*	Mouse	Nuclear	Promotes the translation of *COX1* and *ND1* in mouse C2C12 myoblasts [86]
*miR-5787*	Human	Nuclear	Promotes *COX3* translation and is involved in metabolic reprogramming and cisplatin resistance in Cal27 and Scc25 TSCC cells [87]
*miR-2392*	Human	Nuclear	Decreases the mtDNA-encoded ETC complex subunits by repressing polycistronic mtDNA transcription and enhances anaerobic respiration in Cal27 and Scc9 TSCC cells [88]
**Mitochondria-associated miRNAs**			
*miR-338*	Rat	Nuclear	Decreases the expression of COX4 to reduce mitochondrial oxygen consumption in superior cervical ganglia (SCG) neurons [89]
*miR-210*	Human	Nuclear	Modulates mitochondrial function during hypoxia conditions by reducing expression levels of COX10 in HCT116 colon cancer cells [90]
*miR-101*	Human	Nuclear	Inhibits the replication of herpes simplex virus-1 (HSV-1) by downregulating the expression of ATP5B [91]
*miR-663*	Human	Nuclear	Increases the number of ETC complex (I, II, III, and IV) subunits and assembly factors, thereby stabilizing the ETC supercomplexes in MCF-7 breast cancer cells [92]
**tRFs**			
**mt-tRFs**			
*mt-i-tRF ^Glu(UUC)^*	Human	Mitochondrial	Downregulates MPC1 and increases extracellular lactate levels in MELAS cybrid cells [57]
*tiRNA-5^His(GTG)^*	Rat	Nuclear	Decreases MPC1 and is involved in proliferation and insulin secretion of neonatal rat pancreatic β cells [93]
*t* *iRNA-5^Glu(CTC)^*	Rat	Nuclear	Decreases MPC1 and is involved in proliferation and insulin secretion of neonatal rat pancreatic β cells [93]
**Mitochondria-associated tRFs**			
*tiRNA-5^Glu(TTC)^*	Rat	Nuclear	Decreases MPC1 levels and is involved in mitochondrial respiration, proliferation, and insulin secretion of neonatal rat pancreatic β cells [93]
**lncRNA**			
**mt-lncRNAs**			
*MALAT1*	Human	Nuclear	Modulates the CpG methylation of mtDNA and the expression of mtDNA-encoded ETC-related genes in HepG2 HCC cells [18]
*SAMMSON*	Human	Nuclear	Modulates the expression of mtDNA-encoded ETC complex subunits by facilitating mitochondrial localization of p32 in melanoma cells [94,95]
			*SAMMSON* silencing increases the expression of ETC complexes, cellular respiration, and mitochondrial replication in doxorubicin-resistant MCF-7 breast cancer cells [96]
**Mitochondria-associated lncRNAs**			
*Kcnq1ot1*	Mouse	Nuclear	Inhibits *miR-378a* to upregulate the ATP6 expression and complex V activity in murine cardiomyocytes [97]
*Cerox1*	Mouse	Nuclear	Inhibits *miR-488-3p* to enhance the expression of multiple ETC complex I subunits and mitochondrial respiration in N2A neuroblastoma cells [98]
**circRNAs**			
**mt-circRNAs**			
*circPUM1*	Human	Nuclear	Binds to UQCRC2 to modulate the formation of the UQCRC1-UQCRC2 dimer in mitochondrial complex III in KYSE30 and KYSE410 ESCC cells [99]
*SCAR*	Human	Mitochondrial	Binds to ATP5B to shut down mPTP, thus inhibiting mitochondrial ROS output in liver fibroblasts [76]
*mcPGK1*	Human	Mitochondrial	Promotes mitochondrial import of PGK1 and is involved in metabolic reprogramming from OXPHOS to glycolysis in liver TICs [100]

*ND4*, *NADH ubiquinone oxidoreductase core subunit 4*; *ND2*, *NADH ubiquinone oxidoreductase core subunit 2*; ROS, reactive oxygen species; *COX1*, *cytochrome c oxidase subunit 1*; ATP6, ATP synthase F0 subunit 6; IFM, interfibrillar mitochondria; SSM, subsarcolemmal mitochondria; *CYTB*, *cytochrome b*; *ND1*, *NADH ubiquinone oxidoreductase core subunit 1*; *COX3*, *cytochrome c oxidase subunit 3*; TSCC, tongue squamous cell carcinoma; COX4, cytochrome c oxidase subunit 4; COX10, cytochrome c oxidase assembly factor heme A:farnesyltransferase COX10; ATP5B, ATP synthase F1 subunit β; MPC1, mitochondrial pyruvate carrier 1; *MALAT1*, *metastasis-associated lung adenocarcinoma transcript 1*; HCC, hepatocellular carcinoma; *SAMMSON*, *survival-associated mitochondrial melanoma-specific oncogenic non-coding RNA*; *Kcnq1ot1*, *potassium voltage-gated channel subfamily Q member 1 overlapping transcript 1*; *Cerox1*, *cytoplasmic endogenous regulator of oxidative phosphorylation 1*; *circPUM1*, *circular RNA pumilio-1*; UQCRC2, ubiquinol cytochrome *c* reductase core protein 2; UQCRC1, ubiquinol cytochrome *c* reductase core protein 1; ESCC, esophageal squamous cell carcinoma; *SCAR*, *steatohepatitis-associated circRNA ATP5B regulator*; mPTP, mitochondrial permeability transition pore; *mcPGK1*, *mitochondrial circRNA for translocating phosphoglycerate kinase 1*; PGK1, phosphoglycerate kinase 1; TICs, tumor-initiating cells.

## Data Availability

Not applicable.

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
