# Peer review of "Roles of Noncoding RNAs in Regulation of Mitochondrial Electron Transport Chain and Oxidative Phosphorylation"

_ijms, 2023, doi:10.3390/ijms24119414_

Round 1

Reviewer 1 Report

This is a review paper on the ole of non-coding RNAs on mitochondrial electron transfer chain. The review covers a fascinating aspect of mitochondrial biology that is relatively less known, therefore it is a significant contribution to the undestaning of mitochondrial function. The review examines diffrent types of ncRNAs originating from either nucleus or mitochondria and their effects on the respiratory complexes and on nucleear-mitochondrial communication.

Author Response

Thank you for your valuable comments.

Reviewer 2 Report

Dear Authors,

I very much enjoyed reading this review. Congratulations!

I have only a few suggestions which you can find in the attached pdf.

Other:

The quality of figures should be improved. I suggest also submiting the original images separately to avoid loss of quality due to the pdf compression.

In general it is advisable to avoid have such long blocks of text. Try and split in multiple paragraphs.

Best on luck!

Author Response

Dear Authors,

I very much enjoyed reading this review. Congratulations!

I have only a few suggestions which you can find in the attached pdf.

Thank you for your valuable comments. As you suggested by the reviewer in the attached pdf file, we have modified the following points of the manuscript.

Comment #1. please add definitions for all mito nc RNAs. For example the definition of a mitoMIR. you can add another column in table 1 for this so that everything is visible at first glance

Answer #1. As you suggested, we have added a column to Table 1 showing the definitions of mt-ncRNAs, including mitomiRs.

Comment #2. (Regarding the name of inhibitor of miR-762) Provide name bethween parethesis

Answer #2. As you pointed out, we have provided the name of the inhibitor molecule as follows:

“Intriguingly, the administration of an inhibitor (antagomir) of miR-762 ameliorated myocardial ischemia/reperfusion injury in mice.” (lines 236-237 on page 9 of the revised manuscript).

Comment #3. these blocks of text are very long. I would suggest presenting the miRs based on the mito complex they target and split in paragraphs based on this.

Answer #3. Thank you for your comments. As you suggested, we have divided the long blocks in Chapter 3 and 5 into five and three paragraphs, respectively.

Comment #4. (Regarding the words “This report” in line 270 on page 8) there's a different font here

Answer #4. As you pointed out, we have changed the font to Palatino Linotype in these words (line 346 on page 11 of the revised manuscript).

Comment #5. It would be useful to make some refs to this table (Table 2) in the text not only in the conclusion

Answer #5. As you pointed out, in the revised manuscript, we have moved Table 2 to Chapter 3 and have referred to Table 2 throughout the text as follows:

“Numerous mitomiRs have been shown to affect the levels of enzymes encoded by the mitochondrial genome in the ETC complexes (Table 2 and Figure 3).” (lines 199-200 on page 6)

“Mitochondria-associated miRNAs that are located in the cytoplasm modulate mitochondrial function by targeting nuclear genome-encoded mitochondrial transcripts in the cytoplasm (Table 2 and Figure 3).” (lines 282-284 on page 9)

“As tRNA-derived small fragments (tRFs) are recently discovered as regulators of mitochondrial biology, there are only a few reports on the engagement of tRFs in relation to ETC complex regulation (Table 2 and Figure 3).” (lines 322-324 on page 10)

“These mt-lncRNAs play a crucial role in multiple gene regulatory networks, serving as possible epigenetic mediators that coordinate nuclear and mitochondrial functions [123,124] (Table 2 and Figure 3).” (lines 362-365 on page 11)

“Recent studies have indicated that circRNAs can act as scaffolds in the protein complex assembly and some circRNAs regulates mitochondrial functions through direct binding to subunits of ETC complexes and complex V (Table 2 and Figure 3).” (lines 398-400 on page 12)

Comment #6. It would be nice to have a figure (similar to fig1) presenting all the ncRNAs tatgeting the mito complexes/metabolic processes.

Answer #6. As you suggested, we have added a new figure as Figure 3 presenting all the ncRNAs that are introduced in this review, and have referenced this figure in the sentences as listed in Answer #5 and in the Conclusions section (line 435 on page 12 of the revised manuscript).

Comment #7. (Regarding Table 2) I would color these lines so that they stand out

Answer #7. Thank you for your comments. In the revised manuscript, we have colored the lines indicating the ncRNA categories in Table2.

In addition to the comments above, as you suggested, we have corrected grammatical errors as follows:

“…is the miR-181c that binds to…” (lines 177-178 on page 5 of the original manuscript)

“…is miR-181c that binds to…” (line 238 on page 9 of the revised manuscript)

“…and suppresses the COX1 translation…” (line 179 on page 5 of the original manuscript)

“…and suppresses COX1 translation…” (lines 239-240 on page 9 of the revised manuscript)

“…while, previously, the hypomodification at important sites…” (lines 272-273 on page 7 of the original manuscript)

“…while the hypomodification at important sites…” (lines 348-349 on page 11 of the revised manuscript)

Other:

 Comment #8. The quality of figures should be improved. I suggest also submiting the original images separately to avoid loss of quality due to the pdf compression.

Answer #8. Thank you for your suggestions. We have improved the quality of figures in the revised manuscript and have submitted the original image files separately from the manuscript.

Comment #9. In general it is advisable to avoid have such long blocks of text. Try and split in multiple paragraphs.

Answer #9. Thank you for your comments. As we replied in Answer #3, we have divided long blocks in Chapter 3 and 5 to multiple paragraphs.

Best on luck!

Thank you for your comments.

Reviewer 3 Report

The review by Kobayashi and co-authors focuses on the role of non-coding RNAs in mammalian mitochondria.  This is a novel and exciting aspect of mitochondrial gene expression and OXPHOS function regulation. The authors compile a comprehensive list of non-coding RNAs detected in mitochondria that have been proposed to play a role in modulating respiratory capacity. However, in most cases the molecular mechanisms by which ncRNAs affect mitochondrial function remain largely uncharacterized, making me doubtful of the use of the term “mechanisms” in the review title.

The following points should be addressed:

A central open question remaining regards the import of nuclear encoded RNA into mammalian mitochondria, a subject of intense debate in the field. Some discussion on this aspect should be included in the review.

Page 4, it has been well established that the RNA primer required for mtDNA heavy strand replication is the product of premature arrest of RNA polymerase. Moreover, only a ~130 bp fragment of RMRP has been detected in mitochondria. This portion would not acquire the secondary structure required for function and it is unlikely to act as a nuclease.

Page 6, the sentence stating that COX10 is a subunit of cytochrome c oxidase is incorrect and should be amended. COX10 is not a subunit of complex IV, but a CIV assembly factor required for heme A biosynthesis. 

Table 1, in my opinion could be useful to add in the table the definition and/or length of the different ncRNA types.

minor editing

Author Response

The review by Kobayashi and co-authors focuses on the role of non-coding RNAs in mammalian mitochondria.  This is a novel and exciting aspect of mitochondrial gene expression and OXPHOS function regulation. The authors compile a comprehensive list of non-coding RNAs detected in mitochondria that have been proposed to play a role in modulating respiratory capacity. However, in most cases the molecular mechanisms by which ncRNAs affect mitochondrial function remain largely uncharacterized, making me doubtful of the use of the term “mechanisms” in the review title.

Thank you for your comments. As you suggested, we have changed the title to “Roles of noncoding RNAs in regulation of mitochondrial electron transport chain and oxidative phosphorylation”.

The following points should be addressed:

Comment #1. A central open question remaining regards the import of nuclear encoded RNA into mammalian mitochondria, a subject of intense debate in the field. Some discussion on this aspect should be included in the review.

Answer #1. Thank you for your valuable comments. As you suggested, we have added the following sentences to the text.

“Regarding targeting and import of nDNA-encoded ncRNAs into mammalian mitochondria, some proteins have been indicated to be involved in these processes. For example, polynucleotide phosphorylase (PNPASE) has been suggested to modulate the import of some ncRNAs, including RNase P RNA and RMRP. However, the targeting and import mechanisms of nDNA-encoded ncRNAs into mitochondria remain controversial, and thus new techniques are needed to precisely analyze nuclear-mitochondrial RNA dynamics (reviewed in [64]).” (lines 170-176 on pages 4-5 in the revised manuscript)

Comment #2. Page 4, it has been well established that the RNA primer required for mtDNA heavy strand replication is the product of premature arrest of RNA polymerase. Moreover, only a ~130 bp fragment of RMRP has been detected in mitochondria. This portion would not acquire the secondary structure required for function and it is unlikely to act as a nuclease.

Answer #2. Thank you for your valuable comments. As you indicated, we have modified the text as follows:

“This lncRNA was identified as a component of the mitochondrial RNA processing endoribonuclease (MRP) and has been suggested to be involved in the generation of primers for mitochondrial DNA replication [63]. Meanwhile, the role of RMRP in mitochondria is debated, as recent studies have suggested that RNA primer generation is mediated by premature arrest of RNA polymerase mitochondrial (POLRMT), and the 3’ half of this lncRNA (~130 nt) has been predominantly detected inside mitochondria [64,65].” (lines 151-157 on page 4 of the revised manuscript)

Comment #3. Page 6, the sentence stating that COX10 is a subunit of cytochrome c oxidase is incorrect and should be amended. COX10 is not a subunit of complex IV, but a CIV assembly factor required for heme A biosynthesis.

Answer #3. We appreciate your valuable comments. As you pointed out, we have modified the text as follows:

“…directly reducing expression levels of cytochrome c oxidase subunit 10 (COX10), a subunit of complex IV,” (lines 222-223 on page 6 in the original manuscript)

“…directly reducing expression levels of cytochrome c oxidase assembly factor heme A:farnesyltransferase COX10 that facilitates the assembly of complex IV,” (lines 291-293 on page 10 in the revised manuscript)

Comment #4. Table 1, in my opinion could be useful to add in the table the definition and/or length of the different ncRNA types.

Answer #4. As you suggested, we have added information about length of ncRNAs to Table 1.

Reviewer 4 Report

1. Can you change the title or improve the clarity of the title?

2. Figures are nicely drawn.

3. Page 2: Recent studies, including next generation...... - can you please cite?

4. Can you include how these ncRNAs play a key role in many pathological conditions? 

English is good in this article.

Author Response

Comment #1. Can you change the title or improve the clarity of the title?

Answer #1. Thank you for your comments. As you suggested, we have changed the title to “Roles of noncoding RNAs in regulation of mitochondrial electron transport chain and oxidative phosphorylation”.

Comment #2. Figures are nicely drawn.

Answer #2. Thank you for your comment.

Comment #3. Page 2: Recent studies, including next generation...... - can you please cite?

Answer #3. As you suggested, we have cited the following articles in lines 69-80 on page 3 in the revised manuscript:

New ref. #17.

Das, S.; Ferlito, M.; Kent, O. A.; Fox-Talbot, K.; Wang, R.; Liu, D.; Raghavachari, N.; Yang, Y.; Wheelan, S. J.; Murphy, E.; Steenbergen, C. Nuclear miRNA regulates the mitochondrial genome in the heart. Circ. Res. 2012, 110, 1596–1603

New ref. #18.

Zhao, Y.; Zhou, L.; Li, H., Sun, T.; Wen, X.; Li, X.; Meng, Y.; Li, Y.; Liu, M.; Liu, S.; Kim, S.J.; Xiao, J.; Li, L.; Zhang, S.; Li, W.; Cohen, P.; Hoffman, A.R.; Hu, J.F.; Cui, J. Nuclear-Encoded lncRNA MALAT1 Epigenetically Controls Metabolic Reprogramming in HCC Cells through the Mitophagy Pathway. Mol. Ther. Nucleic Acids. 2020 23, 264-276.

New ref. #19.

Hyttinen, J.M.T.; Blasiak, J.; Kaarniranta, K. Non-Coding RNAs Regulating Mitochondrial Functions and the Oxidative Stress Response as Putative Targets against Age-Related Macular Degeneration (AMD). Int. J. Mol. Sci. 2023 24, 2636.

New ref. #20.

Kamada, S.; Takeiwa, T.; Ikeda, K.; Horie-Inoue, K.; Inoue, S.  Long Noncoding RNAs Involved in Metabolic Alterations in Breast and Prostate Cancers. Front. Oncol. 2020, 10, 593200.

Comment #4. Can you include how these ncRNAs play a key role in many pathological conditions?

Answer #4. As you suggested, we have modified the text as follows:

“Recent studies, including next-generation sequencing strategies, have reported that noncoding RNAs (ncRNAs) play key roles in mitochondrial functions, such as ETC and OXPHOS, through various biological processes including epigenetic and post-transcriptional gene regulation, and are involved in mitochondria- and metabolism-related diseases. For example, among microRNAs (miRNAs), a type of noncoding RNA, miR-181c has been reported to be associated with reduced exercise capacity and cardiac dysfunction by inhibiting translation of cytochrome c oxidase subunit 1 (COX1) and reducing complex IV function [17]. In another type of ncRNAs, long ncRNAs (lncRNAs), metastasis associated lung adenocarcinoma transcript 1 (MALAT1) has been suggested to promote hepatocellular carcinoma (HCC) cell proliferation, migration and invasion by epigenetically regulating mtDNA-encoded ETC-related genes [18]. These ncRNAs may be promising targets in diagnosis and treatment of the diseases [19,20]." (lines 69-80 on page 3 of the revised manuscript)

New ref. #17.

Das, S.; Ferlito, M.; Kent, O. A.; Fox-Talbot, K.; Wang, R.; Liu, D.; Raghavachari, N.; Yang, Y.; Wheelan, S. J.; Murphy, E.; Steenbergen, C. Nuclear miRNA regulates the mitochondrial genome in the heart. Circ. Res. 2012, 110, 1596–1603

New ref. #18.

Zhao, Y.; Zhou, L.; Li, H., Sun, T.; Wen, X.; Li, X.; Meng, Y.; Li, Y.; Liu, M.; Liu, S.; Kim, S.J.; Xiao, J.; Li, L.; Zhang, S.; Li, W.; Cohen, P.; Hoffman, A.R.; Hu, J.F.; Cui, J. Nuclear-Encoded lncRNA MALAT1 Epigenetically Controls Metabolic Reprogramming in HCC Cells through the Mitophagy Pathway. Mol. Ther. Nucleic Acids. 2020 23, 264-276.

New ref. #19.

Hyttinen, J.M.T.; Blasiak, J.; Kaarniranta, K. Non-Coding RNAs Regulating Mitochondrial Functions and the Oxidative Stress Response as Putative Targets against Age-Related Macular Degeneration (AMD). Int. J. Mol. Sci. 2023 24, 2636.

New ref. #20.

Kamada, S.; Takeiwa, T.; Ikeda, K.; Horie-Inoue, K.; Inoue, S.  Long Noncoding RNAs Involved in Metabolic Alterations in Breast and Prostate Cancers. Front. Oncol. 2020, 10, 593200.

“The experiments using spontaneous hypertensive rats have suggested that this mitomiR reduces blood pressure and suppresses cardiac hypertrophy [85].” (lines 262-264 on page 9 of the revised manuscript)

New ref. #85.

Li, H.; Zhang, X.; Wang, F.; Zhou, L.; Yin, Z.; Fan, J.; Nie, X.; Wang, P.; Fu, X.D.; Chen, C.; et al. MicroRNA-21 Lowers Blood Pressure in Spontaneous Hypertensive Rats by Upregulating Mitochondrial Translation. Circulation 2016, 134, 734–751.

MALAT1 depletion impairs the mitochondrial biogenesis and intracellular ATP synthesis, suppresses HCC cell proliferation, migration and invasion, and induces the mitochondria-associated apoptosis. Thus, MALAT1 may be an important therapeutic target for HCC, and the significance of MALAT1 function in mitochondria requires further study [18].” (lines 373-377 on page 11 of the revised manuscript)

Round 2

Reviewer 2 Report

My dears,

Thank you for considering my comments. The new figure is lovely!

Best of luck!

Reviewer 3 Report

The authors have addressed previous comments and in my opinion the manuscript in its current form is suitable for publication.